# Rapidly progressive arthropathy identified on imaging of patients treated with Crizotinib for ALK-rearranged/ROS1-positive non small cell lung cancer: A retrospective single-center study

Yael Eshet[1]*, Liran Domachevsky[1], Noam Tau[2], Gregory Peters Founshtein[1], Michal Eifer[1], Jair Bar[3], Iris Eshed[2]

1 Department of Nuclear Imaging, Sheba Medical Center, Ramat Gan, Israel, 2 Department of Diagnostic Imaging, Sheba Medical Center, Ramat Gan, Israel, 3 Institute of Oncology, Sheba Medical Center, Ramat Gan, Israel

* Yael.eshet@gmail.com, Yael.eshet@sheba.health.gov.il

## Abstract

### Objectives

Progressive arthropathy was anecdotally described in patients exposed to crizotinib, a receptor tyrosine kinases inhibitor (TKI) used to treat anaplastic lymphoma kinase (ALK) or ROS Proto-Oncogene 1 (ROS1) positive non-small cell lung cancer (NSCLC). We aimed to evaluate the incidence of this adverse effect.

### Methods

We retrospectively evaluated imaging studies of all patients in our institution receiving TKI for ALK-rearranged or ROS1-positive NSCLC (crizotinib, alectinib, lorlatinib, brigatinib).

### Results

Between February 2012 and August 2023, out of a total number of 71 subjects (51% male, 36−88 years old) who received TKI's for ALK-rearranged/ ROS1 positive NSCLC, 34/71 (47%) were exposed at least once to crizotinib treatment, while the other 37/71 (53%) patients received other TKI's. Significantly higher incidence (p = 0.02) of irreversible, progressive arthropathy in one or more joints was detected in 18% (6/34, 95% CI: 0.1–0.26) of crizotinib-treated patients, up to 6 years after treatment initiation. Imaging findings included synovial proliferation and progressive arthropathy in hip and shoulder joints, and vertebral endplate destruction.

### Conclusion

We found progressive arthropathy, mostly painless, in one or more joints or intervertebral spaces of patients receiving crizotinib for NSCLC.

**Data availability statement:** The data that support the findings of this study are available on request from the institutional ethics committee (IRB). Requests for data access should be directed to vhelsinki@sheba.health.gov.il.

**Funding:** The author(s) received no specific funding for this work.

**Competing interests:** Dr. Bar declares receiving consulting fees from Merck Sharp & Dohme, Bristol-Myers Squibb, Roche, AstraZeneca, Novartis, Pfizer, Causalis, Bayer, and Takeda, and research grants from AstraZeneca, Takeda, OncoHost, Pfizer, ImmuneAI, Merck Sharp & Dohme, Roche, and Eli Lilly, all for the institute. The other authors declare no conflict of interest. This does not alter our adherence to PLOS ONE policies on sharing data and materials.

**Abbreviations:** TKI, tyrosine kinase inhibitor; ALK, anaplastic lymphoma kinase; ROS1, ROS Proto-Oncogene 1; NSCLC, non-small cell lung cancer; PET, positron emission tomography; FDG, F18-Fludeoxyglucose; MIP, maximum intensity projection.

## Introduction

Crizotinib is a receptor tyrosine kinase inhibitor (TKI) [1], considered first-line treatment for non-small cell lung cancer (NSCLC) whose tumors are anaplastic lymphoma kinase (ALK) or ROS Proto-Oncogene 1 (ROS1) positive, in centers where alectinib is not available [2,3].

Many crizotinib-related adverse events were reported, including hepatotoxicity, interstitial lung disease (ILD)/pneumonitis, upper respiratory infection, QT interval prolongation, and more [1,4–7].

Rapidly progressive osteoarthritis [8] and vertebral end-plate destruction [9] in patients receiving crizotinib were reported in two case studies. Rapidly progressive osteoarthritis is a rare entity of unknown etiology, first acknowledged by Mavrogenis et.al in 1957 as almost complete lysis of the femoral head within a few months [10]. Following papers found the condition in patients suffering from rheumatoid arthritis [11], degenerative osteoarthritis [12], post intra-articular corticosteroid injection [13] and more recently in clinical trials of patients receiving nerve growth factor (NGF) inhibitors [14,15].

We aimed to investigate the incidence and radiological manifestations of progressive arthropathy in patients receiving crizotinib.

## Methods

Approval from an institutional review board was obtained for the retrospective assessment of the patient's clinical and radiological records. The patient's informed consent was waived due to the study's retrospective nature.

### Patient Cohort

The study cohort included all patients in our institution receiving TKI for ALK-rearranged/ ROS1 positive NSCLC (crizotinib, alectinib, lorlatinib, brigatinib) between February 1st, 2012 and August 1st 2023, as identified by a medical records search engine (MDClone Ltd.). The data were accessed retrospectively between August 2023 and September 2023. Patients without baseline and follow-up imaging and under 18 years old were excluded. Patients' imaging and clinical data were retrospectively extracted from their medical records. The cohort was divided into a study group, which included all patients treated with crizotinib at any line of treatment, and a control group, which included patients treated with other types of TKI's but never crizotinib.

### Image evaluation

We (first author, 15 years' experience in radiology and 9 years in nuclear medicine, last author, 20 years' experience in musculoskeletal imaging) evaluated available hips and shoulders' joints as well as the spinal intervertebral spaces on patients' consecutive chest and abdominal CT and PET-CT. The readers had access to information that could identify individual participants, but were blinded to their clinical symptoms and whether they were treated with crizotinib. The readings were performed independently, and discrepancies were settled by consensus.

Synovial joint arthropathy was defined as the novel appearance of joint space narrowing, intra-articular fluid accumulation, intra-articular ossifications and calcifications, extra-articular bursitis, and bone subluxation or deformation. Spinal arthropathy was defined as the novel appearance of joint space narrowing, end plate sclerosis or erosions and osteophyte formation.

Increased F18-Fludeoxyglucose (FDG) uptake was recorded when available.

## Statistical analysis

We performed statistical analyses to compare outcomes between the group of patients exposed to crizotinib, and the group that was not exposed. Two tailed t-test was used for comparing means between groups and $\chi^2$ test was used to compare proportions between groups for categorical variables.

To determine whether the incidence of progressive arthropathy between the two groups was statistically significant we applied a $\chi^2$ test. Given the relatively small sample size in our study, these results were further validated using Monte Carlo simulation (involving generating 10,000 simulated replications to assess the robustness of the observed differences and mitigate the risk of sampling bias) [16]. To examine an interaction between arthropathy and gender, we repeated $\chi^2$ test in the crizotinib group.

To better assess the incidence of arthritis while considering the competing risk of death, we used a competing risks analysis [17]. The cumulative incidence function (CIF) was used to estimate the probability of arthritis (the event of interest) and death (the competing event). The CIFs were stratified by treatment group (Crizotinib exposure vs. no Crizotinib exposure) and compared using Gray's test. Finally, to approximate the effect size of Crizotinib treatment on the risk of developing arthritis (while accounting for the competing risk of death), we applied Fine-Gray regression modeling, and estimated the subdistribution hazard ratio [18]).

All analyses were performed in R (version 4.4.1).

## Results

We identified 71 subjects who received TKI's for NSCLC, of which 34 (15 females, age 65.62 ± 1.83 years (mean ± standard error of the mean (SEM))) were exposed at least once to crizotinib treatment. No patients were excluded due to lack of imaging or age under 18. In most patients, crizotinib was the first treatment line. The crizotinib group had a longer average follow-up time, of over 9 years (p = 0.02) (Table 1).

Out of the 34 patients treated with crizotinib, 6 (18% (95% CI: 0.1–0.26)) had imaging findings consistent with progressive arthropathy, significantly higher than in the control group [$\chi^2$ (degrees of freedom = 1, N = 71) = 5.03, p = 0.02, validated using Monte Carlo simulation, p = 0.008 (10000 repetitions)].

**Table 1. Study and control patient's characteristics.**

| Variable | Total (n = 71) | Crizotinib (n = 34) | No crizotinib (n = 37) | P-value |
|---|---|---|---|---|
| Age at treatment initiation mean (range) | 63 (36-88) | 66 (45-88) | 61 (36-87) | 0.1 |
| Male gender | 36 (51%) | 19 (56%) | 17 (46%) | 0.4 |
| Crizotinib line of treatment | | | | |
| 1st | | 25 | N/A | |
| 2nd | | 4 | N/A | |
| 3rd | | 5 | N/A | |
| Average follow-up, weeks, mean (range) | 152 (6-484) | 215 (6-652) | 194 (41-408) | 0.5 |
| Tumor with ROS1 mutation | 12 (17%) | 12 (35%) | 0 (0%) | |

ROS1, ROS Proto-Oncogene 1; N/A, not applicable

Gray's test revealed a statistically significant difference in the cumulative incidence of arthritis between groups ($\chi^2 = 5.80$, p = 0.016), indicating that arthritis was more common in the Crizotinib group. No significant difference was found for the cumulative incidence of death between groups ($\chi^2 = 2.23$, p = 0.136).

Fine-Gray regression modeling, accounting for the competing risk of death, estimated a subdistribution hazard ratio (HR) of 57.1 for arthritis in the Crizotinib group compared to the no-Crizotinib group. This suggests a markedly increased cumulative incidence of arthritis associated with Crizotinib exposure. However, due to the limited number of arthritis events and the absence of such events in the control group, the HR should be interpreted cautiously.

In the crizotinib group, the incidence of progressive arthropathy was higher in women (15% (95%CI: 0.08–0.22)) than in men (3% (95%CI: 0.0–0.06)), but this was not significant [$\chi^2$ (degrees of freedom = 1, N = 34) = 2.81, p = 0.09]. Most patients had synovial joint involvement. Three patients had multi-articular disease (either both shoulders or several vertebrae) (Table 2).

Time from crizotinib initiation to first imaging findings of arthropathy varied greatly, from 8 to 398 weeks (mean ± SEM: 209 ± 84 weeks). In two of the patients (patient 1 and patient 5), the findings appeared 32 and 34 weeks after crizotinib cessation, in the others the findings appeared under active treatment (Table 2). The destructive process took 9–123 weeks to stabilize (Fig 1).

In the control group, a single patient receiving alectinib had arthritic changes in the shoulder, both on baseline and on follow-up images, without progressive destruction, probably degenerative. No signs of progressive arthropathy were detected in any other patients in the control group receiving other types of TKI inhibitors.

## Clinical correlation with imaging findings of patients with rapidly progressive arthropathy

In patient 1, left shoulder imaging findings appeared two years after crizotinib treatment initiation, preceding the patient's symptoms (shoulder pain). Arthrocentesis to the shoulder joint revealed an inflammatory process with macrophages and synovial cells. Vertebral imaging findings were noted three years following treatment initiation (Fig 2–5), however, clinical records did not indicate complaints of either cervical or back pain.

Patient 3 presented with shoulder pain 142 weeks post-treatment initiation and was diagnosed with rotator cuff injury (Fig 7) and was treated with intra-articular steroid injections, however, eventually the right shoulder joint was replaced.

Patient 4 had advanced shoulder arthropathy changes but was asymptomatic.

Patients 5 and 6 died shortly after the first imaging findings (13 and 11 weeks, respectively), showing progressive arthropathy of the hip and shoulder joint, respectively. Their main symptoms were likely related to their underlying metastatic disease, not specific to the arthritic condition.

**Table 2. Characteristics of patients with progressive arthropathy after exposure to crizotinib.**

| Patient number | 1 | 2 | 3 | 4 | 5 | 6 |
|---|---|---|---|---|---|---|
| Age (years) | 78 | 76 | 82 | 52 | 64 | 67 |
| Gender | F | F | F | F | M | F |
| Mutation | ROS1 | ALK | ROS1 | ROS1 | ROS1 | ALK |
| Treatment line | 1 | 1 | 3 | 1 | 3 | 1 |
| Arthropathy location | | | | | | |
| Synovial joints | 1 | 1 | 2 | 0 | 1 | 1 |
| Spine | 4 | 0 | 0 | 1 | 0 | 0 |
| Total weeks on crizotinib | 69 | 484 | 276 | 398 | 14 | 48 |
| Time between treatment initiation and first imaging findings (weeks) | 101 | 361 | 8 | 103 | 48 | 16 |

F, female; M, male; ROS1, ROS Proto-Oncogene 1; ALK, anaplastic lymphoma kinase.

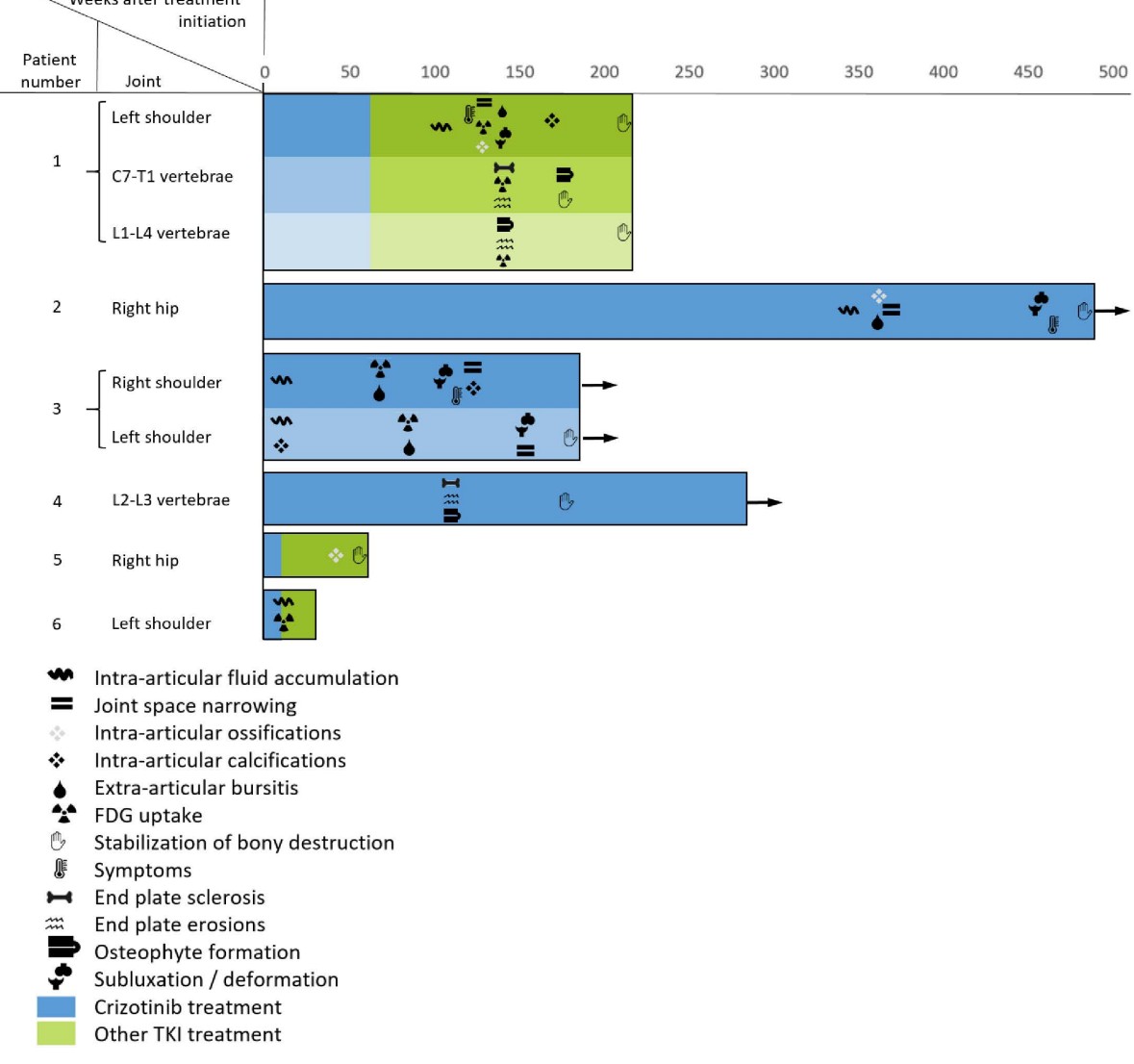

**Fig 1. Swimmer's plot comparing progressive arthropathy imaging findings appearance in all six patients.** Surviving patients at the study's end are shown with arrows.

## Discussion

In this single-center retrospective study, we found that rapidly progressive arthropathy was observed more frequently in one or more joints of patients receiving crizotinib for NSCLC. The imaging findings consisted of synovitis, substantial subluxation and bone deformity in hip and shoulder joints, and vertebral endplate destruction. These were irreversible, and some emerged long after treatment cessation. All spinal destructive changes were asymptomatic. However, synovial joint involvement caused discomfort and disability in the late stages of the process and was commonly misdiagnosed and mistreated – one patient underwent arthrocentesis, another synovectomy, and yet another received a joint replacement. Some patients underwent unnecessary surgical intervention due to synovial joint swelling. Such findings were not found in patients receiving other TKI inhibitors for NSCLC, implying that this is not a class effect but rather may be a drug-specific effect.

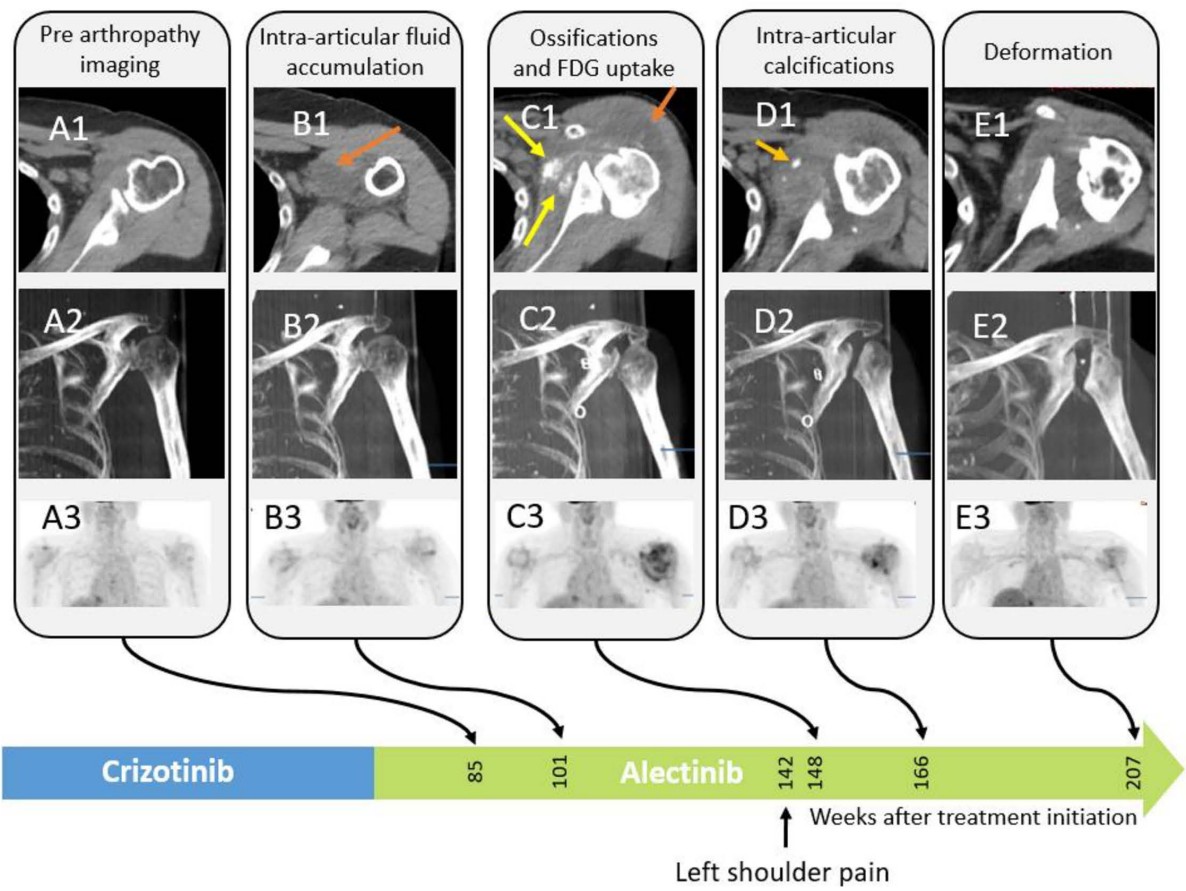

**Fig 2. Images of the left shoulder of a 78-year-old female treated with crizotinib.** – CT and PET MIP images showing progressive articular and bursa fluid accumulation, joint space narrowing, extra articular bursitis intra-articular ossification progressive deformation of the humeral head and gradual increase in FDG uptake compatible with synovitis (arrows).

Findings of progressive osteoarthritis in a patient receiving crizotinib have been described as a case report in a single 75-year-old female patient [8] with ROS1 positive NSCLC, 4.5 years after treatment initiation. Guisier et al [9] reported a case of progressive thoracic vertebral osteitis in a 31-year-old woman with ALK rearranged NSCLC, three months after treatment initiation. Our study also found vast variation in the time span between treatment initiation and findings. Finally, an image of vertebral endplate sclerosis and irregularity in a crizotinib-treated patient was included in a review on targeted therapies and was attributed to osteopenia [19].

Neuropathic arthropathy in lower extremity joints was described in a case report of two patients treated with the NTRK inhibitor entrectinib, a TKI that affects the differentiation and survival of sensory neurons [20]

Nevertheless, progressive arthropathy was not mentioned as a potential adverse reaction to crizotinib in the drug's brochure by Pfizer [1] nor in by a review of TKI's [19]. This may potentially stem from a shorter follow-up time in these cited studies of 10.9 months (48 weeks), while first findings in some of our inflicted patients appeared much later (mean±SEM 106±53 weeks).

Due to the retrospective nature of the study, we can't determine the mechanism of progressive arthropathy found in our study. The characteristic imaging findings of progressive arthropathy are similar to the ones seen in neuropathic joints [21–27]. ALK and ROS1 receptors are expressed in various tissues, including nerve cells and peripheral nerve endings.

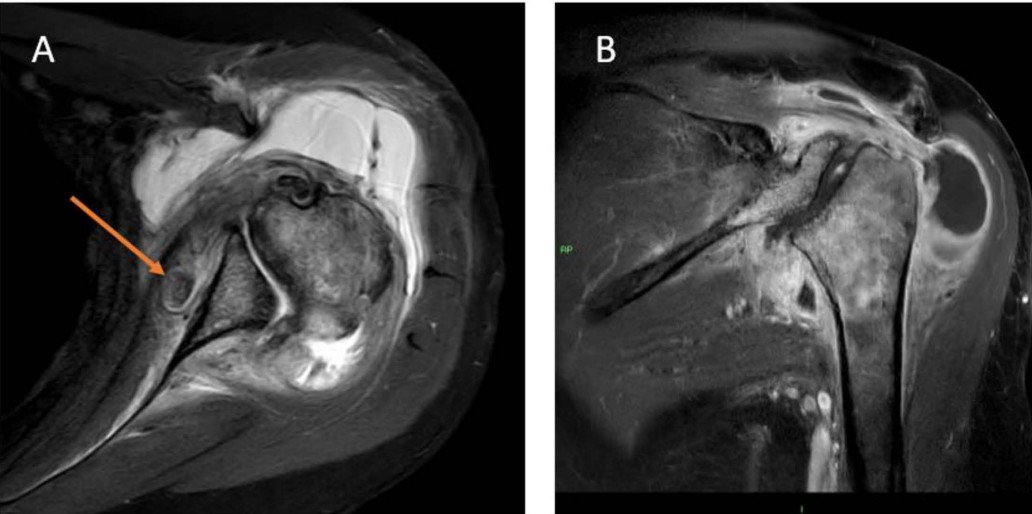

**Fig 3. MRI images of the left shoulder of a 78-year-old female, 151 weeks after crizotinib initiation.** (A) Axial proton density with fat suppression and (B) coronal-oblique T1-weighted images with fat suppression after contrast injection sequences. Images show deformation and subluxation of the humeral head, diffuse bone marrow edema of the proximal humerus and the glenoid, significant amount of intra-articular fluid as well as severe synovitis that enhances after contrast injection. There is also a free intra-articular fragment located anteriorly to the glenoid (arrow).

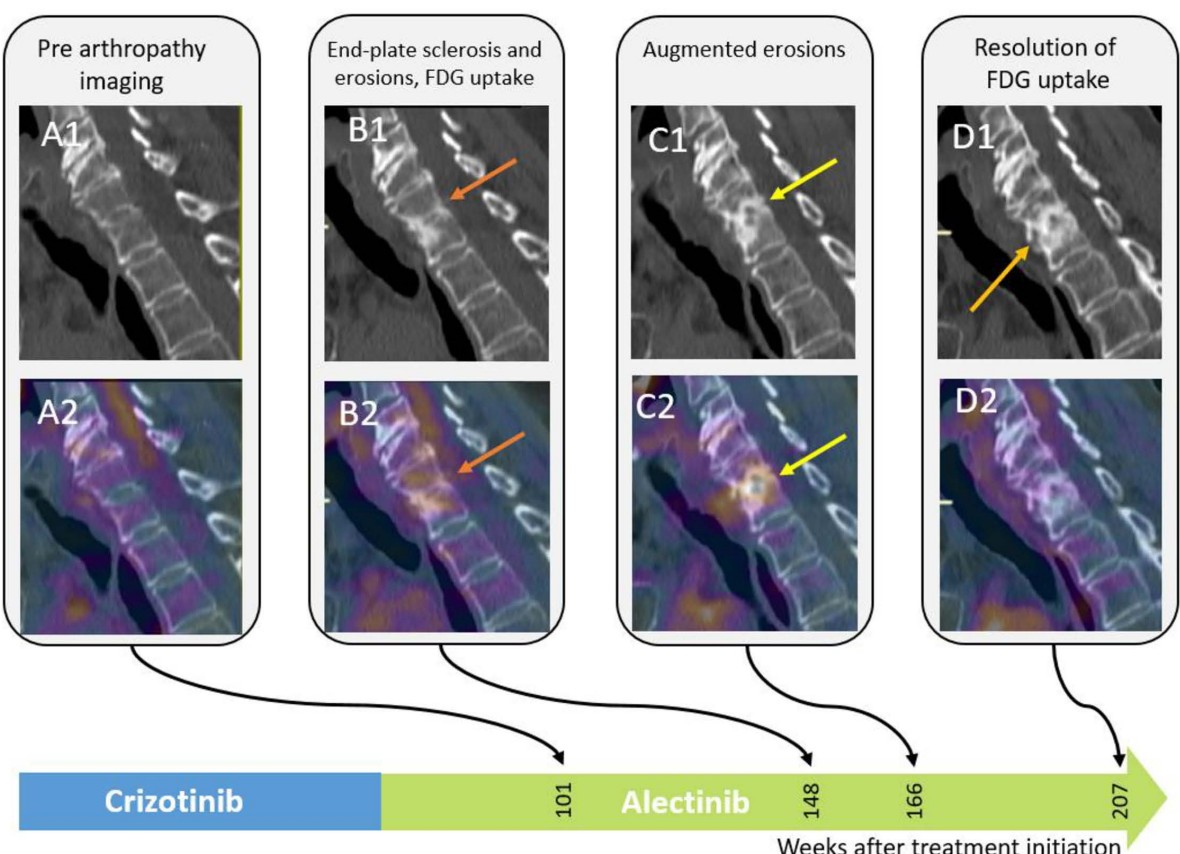

**Fig 4. Images of the cervical spine of a 78-year-old female treated with crizotinib.** Sagittal CT reformats of the cervical spine and Fused PET/CT images show end plate sclerosis, irregularities and erosions with osteophyte formation with increased FDG uptake (arrows).

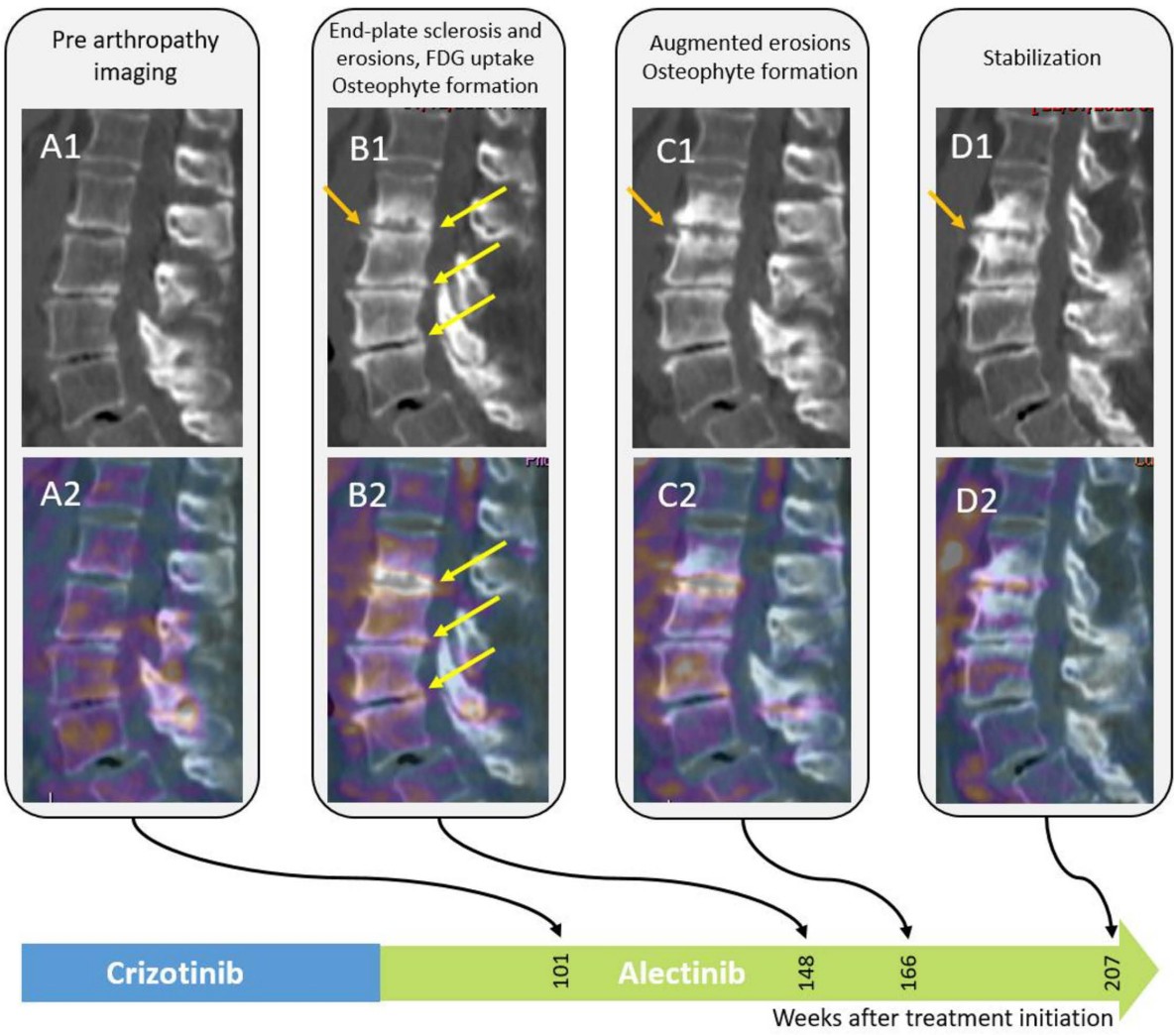

**Fig 5. Images of the lumbar spine of a 78-year-old female treated with crizotinib.** Row 1: Sagittal CT reformats of the lumbar spine and fused PET/CT images show end-plate irregularities and erosions along with osteophyte formation with increased FDG uptake, without soft tissue changes. Patient 2 presented with fever and swollen (albeit non-painful) right hip, 465 weeks after treatment initiation, and underwent synovectomy due to suspected septic arthritis and psoas abscess (Fig 6). Pathology report disclosed inflammatory changes, negative for bacterial infection.

We can hypothesize that crizotinib affects nerve function directly or indirectly through its action on tyrosine kinases in these tissues. This hypothesis may explain the discrepancy between the scant patient's symptoms and the aggressiveness of the imaging findings. In addition, disruption of signalling pathways involved in inflammation and tissue repair, which are influenced by ALK and ROS1 receptors [28] may contribute to joint inflammation and damage, further exacerbating neuropathic arthropathy.

Rapidly progressive osteoarthritis type II was also reported as an adverse event of nerve growth factor inhibitors [14,15], affecting hip, knee and shoulder joints, specifically combined with NSAID's treatment.

The study's inherent limitations include the small cohort and the retrospective evaluation. We were limited to assessing the shoulders, hips, and vertebrae included in chest and abdominal CTs, and thus were not able to evaluate distant joints. In addition, average follow-up time for patients who were exposed to crizotinib was higher than in

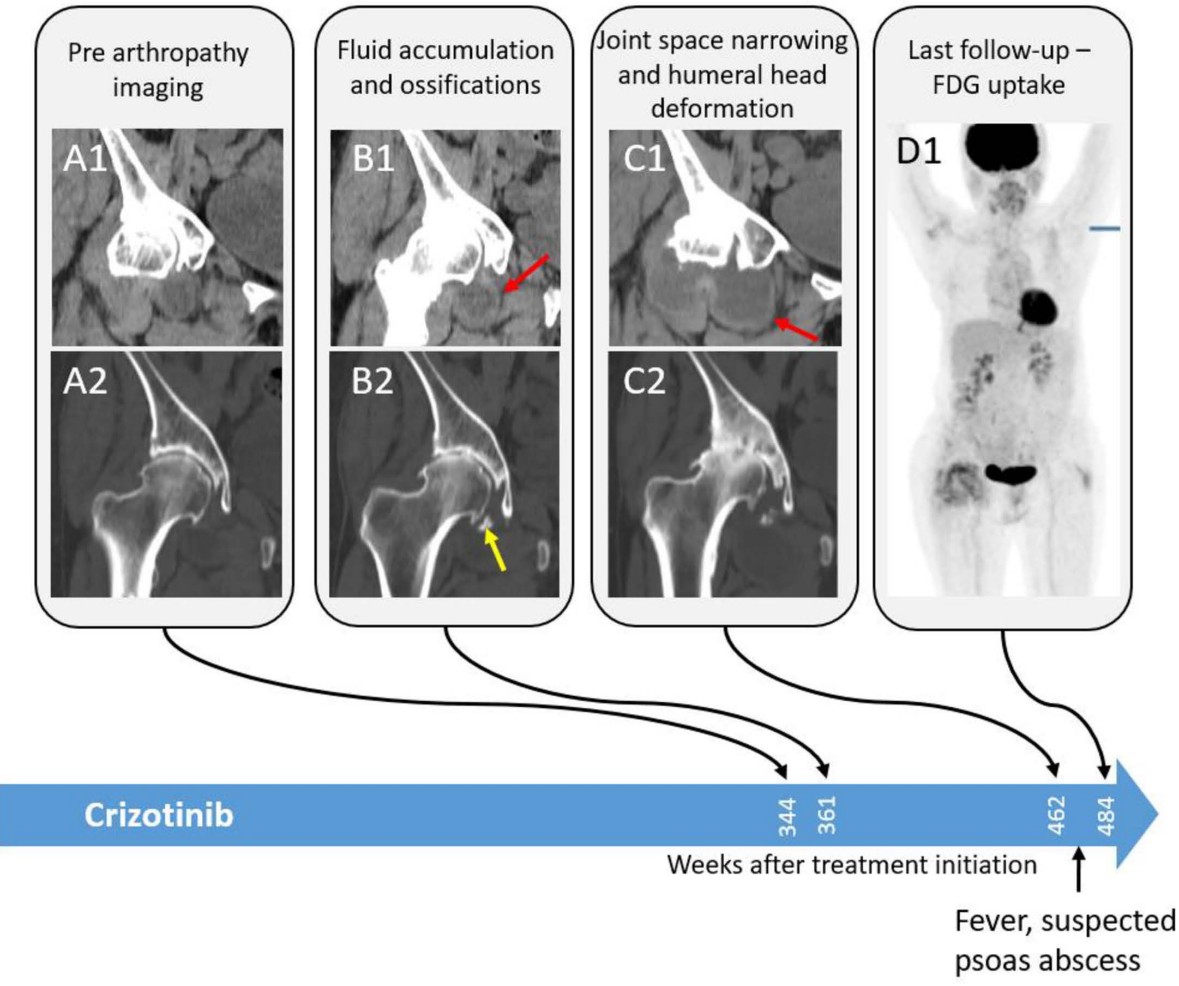

**Fig 6. CT images of progressive arthropathy of the right hip in a 76-year-old female treated with crizotinib.** Soft tissue and bone window coronal CT reformats of the right hip show progressive synovial thickening and intra-articular fluid accumulation in the right hip joint, with narrowing of the joint space, intra-articular ossifications, and new femoral head deformation accompanied by increased diffuse right hip uptake almost 7 years after treatment initiation.

patients without exposure (mean 43 vs. 28 months respectively), probably because crizotinib was the first TKI to be approved by the FDA for ALK-rearranged/ ROS1 positive NSCLC. Finally, as joint destruction typically manifested long after treatment initiation in most cases, this led to a selection bias, excluding patients with shorter survival times. Due to the retrospective nature of the study, assessing the symptoms of the affected patients was limited to the data in the clinical records. There is imbalance between the study and control groups, as only the study group had ROS1 mutation (only crizotinib was reimbursed initially by the HMO for patients with this mutation), and this is a possible confounder.

To conclude, we found progressive arthropathy, mostly painless, in one or more joints or intervertebral spaces of patients receiving crizotinib for NSCLC, which can affect patients' quality of life. Although crizotinib is no longer the first treatment line for most ALK-rearranged/ ROS1-positive NSCLC, it is still in use, and clinicians and radiologists should be aware of this potential adverse event.

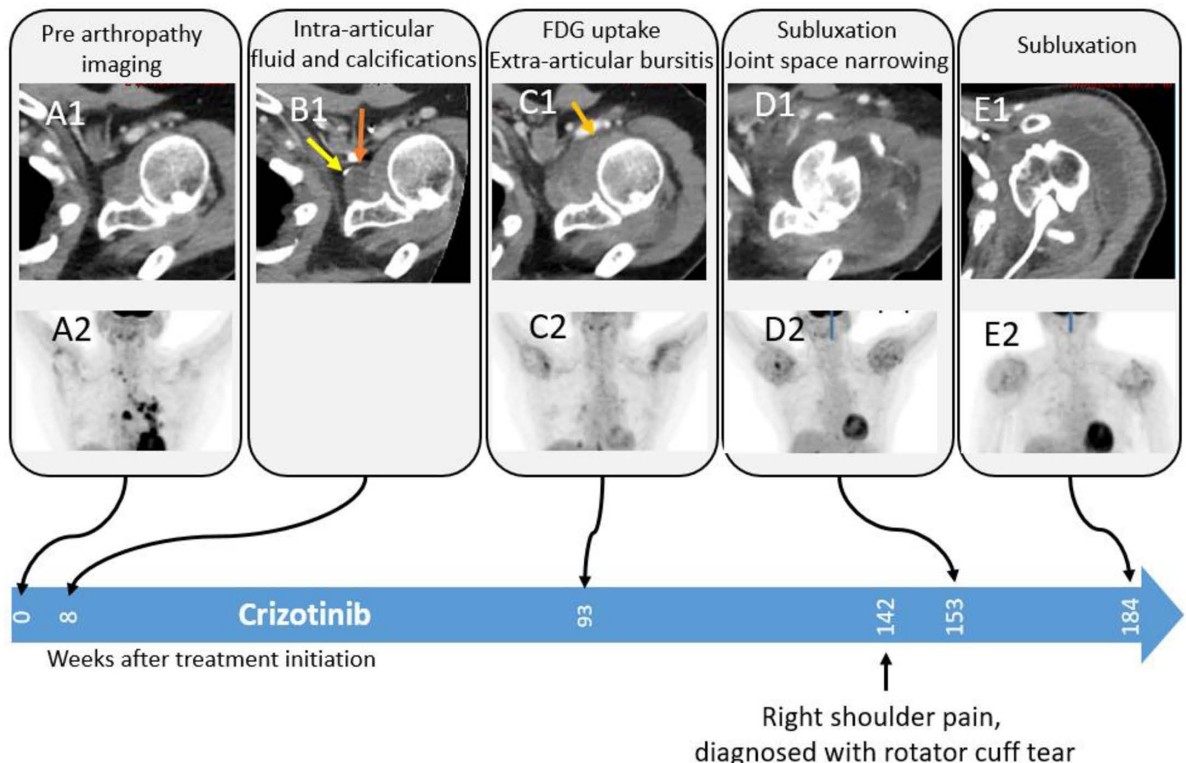

**Fig 7. Progressive arthropathy of the left shoulder in an 82-year-old female treated with crizotinib.** Soft tissue window axial CT and MIP- FDG PET images show progressive articular fluid accumulation in the left shoulder joint and intra-articular calcifications 8 weeks after treatment initiation, followed by hypermetabolic synovial thickening, extra-articular bursitis, joint space narrowing, and humeral head deformation. Finally, humeral head dislocation occurred.

## Author contributions

**Conceptualization:** Yael Eshet, Noam Tau.

**Data curation:** Yael Eshet, Jair Bar.

**Formal analysis:** Yael Eshet, Gregory Peters Founshtein, Iris Eshed.

**Investigation:** Yael Eshet, Iris Eshed.

**Methodology:** Yael Eshet, Noam Tau, Gregory Peters Founshtein, Michal Eifer.

**Project administration:** Yael Eshet.

**Resources:** Liran Domachevsky, Jair Bar.

**Software:** Gregory Peters Founshtein.

**Supervision:** Yael Eshet, Noam Tau.

**Validation:** Liran Domachevsky, Gregory Peters Founshtein, Michal Eifer.

**Writing – original draft:** Yael Eshet, Iris Eshed.

**Writing – review & editing:** Liran Domachevsky, Noam Tau, Gregory Peters Founshtein, Michal Eifer, Jair Bar.

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
