## [Decision Letter · Decision Letter 0]

23 Jun 2025

Dear Dr. Eshet,

Thank you for submitting your manuscript to PLOS ONE. After careful consideration, we feel that it has merit but does not fully meet PLOS ONE’s publication criteria as it currently stands. Therefore, we invite you to submit a revised version of the manuscript that addresses the points raised during the review process.

We look forward to receiving your revised manuscript.

Kind regards,

Avaniyapuram Kannan Murugan, M.Phil., Ph.D.

Academic Editor

PLOS ONE

2. In the online submission form you indicate that your data is not available for proprietary reasons and have provided a contact point for accessing this data. Please note that your current contact point is a co-author on this manuscript. According to our Data Policy, the contact point must not be an author on the manuscript and must be an institutional contact, ideally not an individual. Please revise your data statement to a non-author institutional point of contact, such as a data access or ethics committee, and send this to us via return email. Please also include contact information for the third party organization, and please include the full citation of where the data can be found.

“Dr. Bar declares receiving consulting fees from Merck Sharp & Dohme, Bristol-Myers Squibb, Roche, AstraZeneca, Novartis, Pfizer, Causalis, Bayer, and Takeda, and research grants from AstraZeneca, Takeda, OncoHost, Pfizer, ImmuneAI, Merck Sharp & Dohme, Roche, and Eli Lilly, all for the institute.

The other authors declare no conflict of interest.”

Additional Editor Comments:

The study is very interesting and accordingly reviewers comments also reflected. However, reviewers raise many concern to address before consideration for publication. Address all the comments thoroughly point-by-point manner. Cite these potential articles (PMID: 21596819; PMID: 34462665).

Reviewers' comments:

Reviewer's Responses to Questions

**Comments to the Author**

1. Is the manuscript technically sound, and do the data support the conclusions?

Reviewer #1: Yes

Reviewer #2: Yes

2. Has the statistical analysis been performed appropriately and rigorously?

Reviewer #1: Yes

Reviewer #2: No

3. Have the authors made all data underlying the findings in their manuscript fully available?

Reviewer #1: Yes

Reviewer #2: No

4. Is the manuscript presented in an intelligible fashion and written in standard English?

Reviewer #1: Yes

Reviewer #2: Yes

Reviewer #1: The study addresses a clinically important, underreported adverse event associated with crizotinib use. The design is appropriate, the imaging data is interesting, and the results could raise awareness in clinical practice. However, several important corrections and improvements are necessary before publication. These include:

1. The abstract needs to mention the total number of patients and compare incidence rates between the crizotinib and non-crizotinib groups more explicitly.

2. Correct this typo (e.g., "TKI's" instead of "TKIs"). Language polishing is needed to improve readability.

3. The flow from introduction to methods could be more logically ordered (e.g., first describe the cohort, then imaging evaluation, then analysis).

4. The use of the χ² test and Monte Carlo validation is appropriate, but needs a clearer explanation in the Methods.

5. Confidence intervals (CI) for key results should be reported alongside p-values for incidence of arthropathy (e.g., 18% (95% CI: x–y%)).

6. The authors mention a longer follow-up time for crizotinib-treated patients but don't adjust statistically for it; suggest a time-to-event analysis (Kaplan-Meier) if feasible.

7. The clinical symptoms were often mild or absent—this limits the impact of imaging findings alone. The authors should discuss this discrepancy more critically: How clinically significant is the arthropathy?

8. Table 1 and Table 2 formatting should be improved (e.g., proper alignment of columns, p-values indicated).

9. The crizotinib treatment time row in Table 1 lacks corresponding column for the non-crizotinib group. Also, what about the female gender information in Table 1?

10. The reference list is short; the author should cite more works to enhance its clinical relevance.

The study is novel and potentially important, but requires substantial revisions for clarity, robustness, and adherence to journal standards.

Reviewer #2: Manuscript Title:

Patients treated with Crizotinib for ALK-rearranged/ROS1-positive non-small cell lung cancer: a retrospective single-center study

Summary and General Comments

This is a retrospective observational study that evaluates the incidence and imaging characteristics of progressive arthropathy in ALK/ROS1-positive NSCLC patients treated with crizotinib, compared to those treated with other TKIs. The authors report a significantly higher incidence of progressive joint changes (18%, 6/34) among crizotinib-treated patients.

This is a novel and clinically relevant observation that is not well described in the current literature. The imaging examples provided are clear and compelling. The study addresses an under-recognized adverse effect that could influence radiologic and clinical practice.

However, the study is limited by small sample size, control group imbalance, potential survival bias, and lack of statistical adjustment for key confounders (mutation status, follow-up time). As written, the data suggest an association between crizotinib exposure and progressive arthropathy, but do not definitively establish a drug-specific causal effect.

The authors’ call for increased awareness of this potential adverse effect is reasonable, but the strength of the conclusions should be tempered to reflect the limitations of the retrospective design and small cohort.

Several methodological clarifications and adjustments are needed to strengthen the rigor and transparency of the manuscript:

Major Comments

1. Control group imbalance and confounding

From Table 1:

• ROS1 mutation is present in 35% of the crizotinib group vs. 0% in the control group.

• Average follow-up time is 185 weeks (crizotinib) vs. 121 weeks (controls), p = 0.02.

No adjustment is made for this difference. Since ROS1-positive patients may have longer survival, and longer follow-up gives more chance for late toxicity to appear, the reported 18% arthropathy incidence may be inflated compared to the control group.

Add either a Cox regression or multivariate logistic regression adjusting for follow-up duration and mutation type, or, at minimum, discuss this limitation clearly in the Discussion.

Note that while the authors validate the chi-squared p-value using Monte Carlo simulation (p = 0.008), this does not address the more important issue of group imbalance and survival bias.

2. Definition and reliability of imaging assessment

The definition of “progressive arthropathy” relies on a set of imaging findings, but the assessment process is not clearly described:

• Were the radiologists blinded to treatment group?

• Were readings performed independently or by consensus?

• Was any inter-rater reliability calculated (e.g., kappa)?

Please clarify these important methodological points. Without blinding or assessment of agreement, there is potential for observer bias.

3. Timing of toxicity onset vs. exposure

In several cases, arthropathy occurred long after treatment initiation (e.g., 361 or 465 weeks), and it is not stated if the patients were still receiving crizotinib at the time of onset.

Please clarify how many of the six patients were still on crizotinib when arthropathy first appeared, and for any patients off treatment, how long after discontinuation the findings were detected.

This will inform the clinical interpretation of whether this is an on-treatment or delayed toxicity.

4. Clinical impact of arthropathy

While the imaging findings are clear, the clinical relevance is not well detailed:

• How many patients experienced pain, mobility limitations, or required joint replacement or other interventions?

• What was the impact on quality of life or functional status?

Adding this clinical data will greatly enhance the relevance of the findings.

5. Selection bias and survival bias

As acknowledged by the authors, patients with shorter survival were unlikely to develop late arthropathy, resulting in survival bias.

However, this issue is not sufficiently addressed in the interpretation of the results: the conclusion currently states “There is a higher incidence…” which overstates the certainty.

Please modify this conclusion to reflect the limitations (e.g., “may be associated with higher risk,” or “was observed more frequently in this retrospective cohort, though subject to bias”).

Minor Comments

1. Figure legends are overly descriptive and read like clinical notes. Please revise for clarity and conciseness — e.g., “Representative CT images of progressive shoulder arthropathy after crizotinib treatment.” The legends currently include long lists of arrows (red arrow, yellow arrow, orange arrow- which are not seen until Fig 3) and serial descriptions of individual images. For example, Figure 2a: please check the figures for this

“Row 1: A1 – Pre-arthropathy CT image. B1 - progressive articular and bursa fluid accumulation (red arrow)…”

Please revise for clarity and conciseness, following the style of standard figure legends. For example:

“Representative CT images of progressive shoulder arthropathy following crizotinib treatment, showing synovial thickening, joint space narrowing, and bony deformity.”

Similar revisions are suggested for Figures 2b, 2c, 2d, 3, and 4.

2. Repetitive phrasing in Discussion opening paragraph: “progressive arthropathy in one or more joints... progressive arthropathy leading to substantial subluxation...” suggest rephrasing to improve flow.

3. Tense inconsistencies: e.g., “Findings of progressive osteoarthritis were so far described…” should be “have been described...”

4. Table 1 layout: suggest reformatting with columns: Variable | Total (n=71) | Crizotinib (n=34) | No Crizotinib (n=37) | P-value

5. Mechanistic hypothesis: The discussion about possible neuropathic mechanisms should be framed more explicitly as hypothesis, given the retrospective nature of the study.

Recommendation

Minor revision

This is a strong and clinically relevant study. The imaging data is valuable and adds important knowledge to the field. The methodological concerns, control group imbalance, confounding, survival bias, and definition of assessment, can be addressed through clarifications, more transparent reporting, and minor statistical adjustments.

**Do you want your identity to be public for this peer review?** For information about this choice, including consent withdrawal, please see our Privacy Policy

Reviewer #1: No

Reviewer #2: No

---

## [Author Response · Author response to Decision Letter 1]

4 Jul 2025

Dear Prof. Murugan,

Academic Editor

PLOS ONE

Thank you for the opportunity to submit a revised draft of my manuscript titled “Rapidly progressive arthropathy identified on imaging of patients treated with Crizotinib for ALK-rearranged/ROS1-positive non-small cell lung cancer: a retrospective single-center study” to PLOS ONE.

We appreciate the time and effort you and the reviewers have dedicated to providing valuable feedback on our manuscript. We are grateful for their insightful comments on our paper. We have incorporated changes to address the reviewers' suggestions. The changes have been highlighted within the manuscript.

Here is a point-by-point response to the reviewers’ comments and concerns.

Comments from Reviewer #1

Comment 1:

The abstract needs to mention the total number of patients and compare incidence rates between the crizotinib and non-crizotinib groups more explicitly.

Response:

Thank you for this valuable comment. We have revised the results section of the abstract to emphasize the total number of patients and the incidence rates in each group:

"Between February 2012 and August 2023, out of a total number of 71 subjects (51% male, 36-88 years old) who received TKI’s for ALK-rearranged / ROS1 positive NSCLC, 34/71 (47%) were exposed at least once to crizotinib treatment, while the other 37/71 (53%) patients received other TKI's. "

Comment 2:

Correct this typo (e.g., "TKI's" instead of "TKIs"). Language polishing is needed to improve readability.

Response:

We appreciate you pointing this out. We have corrected TKIs to TKI's in both the patient cohort and the discussion sections.

Comment 3:

The flow from introduction to methods could be more logically ordered (e.g., first describe the cohort, then imaging evaluation, then analysis).

Response:

Thank you for this suggestion. The flow is as requested in the methods section: patient cohort, image evaluation and then analysis.

Comment 4:

The use of the χ² test and Monte Carlo validation is appropriate, but needs a clearer explanation in the Methods.

Response:

Thank you for this important comment. We have clarified how each analysis contributed to the study’s objectives and included reasoning for the Monte Carlo simulation, addressing the small sample size.

In the methods section, we added:

“We performed statistical analyses to compare outcomes between the group of patients exposed to crizotinib, and the group that was not exposed. Two tailed t-test was used for comparing means between groups and χ2 test was used to compare proportions between groups for categorical variables.

To determine whether the incidence of progressive arthropathy between the two groups was statistically significant we applied a χ2 test. Given the relatively small sample size in our study, these results were further validated using Monte Carlo simulation (involving generating 10,000 simulated replications to assess the robustness of the observed differences and mitigate the risk of sampling bias).”

Comment 5:

Confidence intervals (CI) for key results should be reported alongside p-values for incidence of arthropathy (e.g., 18% (95% CI: x–y%)).

Response:

Thank you for this important comment. Confidence interval values have been added to the incidences:

“Out of the 34 patients treated with crizotinib, 6 (18% (95%CI: 0.1-0.26%)) had imaging findings consistent with progressive arthropathy, significantly higher than in the control group [χ2 (degrees of freedom = 1, N = 71) = 5.03, p = 0.02, validated using Monte Carlo simulation, p = 0.008 (10000 repetitions)].

In the crizotinib group, the incidence of progressive arthropathy was higher in women (15% (95%CI: 0.08-0.22%)) than in men (3% (95%CI: 0.0-0.06%)), but this was not significant [χ2 (degrees of freedom = 1, N = 34) = 2.81, p = 0.09].”

Comment 6:

The authors mention a longer follow-up time for crizotinib-treated patients but don't adjust statistically for it; suggest a time-to-event analysis (Kaplan-Meier) if feasible.

Response:

We appreciate this valuable suggestion. To address the disparity in follow-up duration between groups, we updated the data to the present time, resulting in longer and more comparable follow-up periods for both groups. Since crizotinib has not been the first-line treatment at our institution in recent years, follow-up times have become more balanced. The revised follow-up data are now shown in Tables 1 and 2 in the results section.

As proposed, we added a time-to-event analysis. We used the cumulative incidence function (CIF) instead of the Kaplan-Meier method, as it is more suitable for situations with competing risks, such as death and adverse events.

The following text has been added to the methods section:

“To better assess the incidence of arthritis while considering the competing risk of death, we used a competing risks analysis. The cumulative incidence function (CIF) was used to estimate the probability of arthritis (the event of interest) and death (the competing event). The CIFs were stratified by treatment group (Crizotinib exposure vs. no Crizotinib exposure) and compared using Gray’s test. Finally, to approximate the effect size of Crizotinib treatment on the risk of developing arthritis (while accounting for the competing risk of death), we applied Fine-Gray regression modeling, and estimated the subdistribution hazard ratio).

All analyses were performed in R (version 4.4.1).”

And to the results section:

“Gray’s test revealed a statistically significant difference in the cumulative incidence of arthritis between groups (χ² = 5.80, p = 0.016), indicating that arthritis was more common in the Crizotinib group. No significant difference was found for the cumulative incidence of death between groups (χ² = 2.23, p = 0.136).

Fine-Gray regression modeling, accounting for the competing risk of death, estimated a subdistribution hazard ratio (HR) of 57.1 for arthritis in the Crizotinib group compared to the no-Crizotinib group. This suggests a markedly increased cumulative incidence of arthritis associated with Crizotinib exposure. However, due to the limited number of arthritis events and the absence of such events in the control group, the HR should be interpreted cautiously.”

Comment 7:

The clinical symptoms were often mild or absent—this limits the impact of imaging findings alone. The authors should discuss this discrepancy more critically: How clinically significant is the arthropathy?

Response:

Thank you for raising this excellent point. Because of the retrospective nature of the study, there are missing data regarding patients' symptoms. The clinical records do not consistently document pain. The known symptoms and interventions experienced by each patient are listed separately in the results section. The primary symptoms seemed to be joint swelling, discomfort, and disability, occasionally leading to unnecessary surgical interventions, which we have added to the discussion section:

“One patient underwent arthrocentesis, another synovectomy, and yet another joint replacement.”

And to the limitations section:

“Due to the retrospective nature of the study, assessing the symptoms of the affected patients was limited to the data in the clinical records.”

Comment 8:

Table 1 and Table 2 formatting should be improved (e.g., proper alignment of columns, p-values indicated).

Response:

We appreciate this feedback. The formatting for Tables 1 and 2 has been improved.

Comment 9:

The crizotinib treatment time row in Table 1 lacks corresponding column for the non-crizotinib group. Also, what about the female gender information in Table 1?

Response:

Thank you for these points. The non-crizotinib group was not exposed to crizotinib, and therefore, this information was marked as N/A. To avoid confusion on the matter, the data was deleted from the table.

Female gender statistics are the mirror image of the male gender statistics presented: 49% in the entire group, with 46% of them in the crizotinib group. The table shows no statistical significance between the groups regarding gender (p = 0.4).

Comment 9:

The reference list is short; the author should cite more works to enhance its clinical relevance.

Response:

We agree with your comment. Thirteen references have been added.

Comments from Reviewer #2:

Major Comments

Comment 1:

Control group imbalance and confounding. From Table 1:

• ROS1 mutation is present in 35% of the crizotinib group vs. 0% in the control group.

Response:

Thank you for highlighting this imbalance. The imbalance stems from the fact that the other TKI's (alectinib, lorlatinib, brigatinib) were initially not reimbursed by our HMO for ROS1 mutations.

We have added the following to the limitations section:

“There is imbalance between the study and control groups, as only the study group had ROS1 mutation (only crizotinib was reimbursed initially by the HMO for patients with this mutation), and therefore this is a possible confounder.”

Comment 2:

Average follow-up time is 185 weeks (crizotinib) vs. 121 weeks (controls), p = 0.02.

No adjustment is made for this difference. Since ROS1-positive patients may have longer survival, and longer follow-up gives more chance for late toxicity to appear, the reported 18% arthropathy incidence may be inflated compared to the control group.

Add either a Cox regression or multivariate logistic regression adjusting for follow-up duration and mutation type, or, at minimum, discuss this limitation clearly in the Discussion.

Note that while the authors validate the chi-squared p-value using Monte Carlo simulation (p = 0.008), this does not address the more important issue of group imbalance and survival bias.

Response:

We appreciate your detailed feedback on this point.

To address the disparity in follow-up duration between groups, we updated the data to the present time, resulting in longer and more comparable follow-up periods for both groups. Since crizotinib has not been the first-line treatment at our institution in recent years, follow-up times have become more balanced. The revised follow-up data are now shown in Tables 1 and 2 in the results section.

As proposed, we added a time-to-event analysis. We used the cumulative incidence function (CIF) instead of the Kaplan-Meier method, as it is more suitable for situations with competing risks, such as death and adverse events.

The following text has been added to the methods section:

“To better assess the incidence of arthritis while considering the competing risk of death, we used a competing risks analysis. The cumulative incidence function (CIF) was used to estimate the probability of arthritis (the event of interest) and death (the competing event). The CIFs were stratified by treatment group (Crizotinib exposure vs. no Crizotinib exposure) and compared using Gray’s test. Finally, to approximate the effect size of Crizotinib treatment on the risk of developing arthritis (while accounting for the competing risk of death), we applied Fine-Gray regression modeling, and estimated the subdistribution hazard ratio).

All analyses were performed in R (version 4.4.1).”

And to the results section:

“Gray’s test revealed a statistically significant difference in the cumulative incidence of arthritis between groups (χ² = 5.80, p = 0.016), indicating that arthritis was more common in the Crizotinib group. No significant difference was found for the cumulative incidence of death between groups (χ² = 2.23, p = 0.136).

Fine-Gray regression modeling, accounting for the competing risk of death, estimated a subdistribution hazard ratio (HR) of 57.1 for arthritis in the Crizotinib group compared to the no-Crizotinib group. This suggests a markedly increased cumulative incidence of arthritis associated with Crizotinib exposure. However, due to the limited number of arthritis events and the absence of such events in the control group, the HR should be interpreted cautiously.”

Comment 2:

Definition and reliability of imaging assessment

The definition of “progressive arthropathy” relies on a set of imaging findings, but the assessment process is not clearly described:

• Were the radiologists blinded to treatment group?

• Were readings performed independently or by consensus?

• Was any inter-rater reliability calculated (e.g., kappa)?

Please clarify these important methodological points. Without blinding or assessment of agreement, there is potential for observer bias.

Response:

Thank you for pointing out these important methodological considerations. We have added to the methods section:

“The readers had access to information that could identify individual participants, but were blinded to the treatment groups. The readings were performed independently, and discrepancies were settled by consensus. “

Comment 3:

Timing of toxicity onset vs. exposure

In several cases, arthropathy occurred long after treatment initiation (e.g., 361 or 465 weeks), and it is not stated if the patients were still receiving crizotinib at the time of onset.

Please clarify how many of the six patients were still on crizotinib when arthropathy first appeared, and for any patients off treatment, how long after discontinuation the findings were detected.

This will inform the clinical interpretation of whether this is an on-treatment or delayed toxicity.

Response:

We appreciate you asking for this important clarification. This information has been added to the results section:

“In two of the patients (patient 1 and patient 5), the findings appeared 32 and 34 weeks after crizotinib cessation, in the others the findings appeared under active treatment (table 2).”

Comment 4:

Clinical impact of arthropathy

While the imaging findings are clear, the clinical relevance is not well detailed:

• How many patients experienced pain, mobility limitations, or required joint replacement or other interventions?

• What was the impact on quality of life or functional status?

Adding this clinical data will greatly enhance the relevance of the findings.

Response:

Thank you for this insightful comment. Due to the retrospective nature of the study, and since most patients experienced little pain related to the arthropathy, there is limited data regarding patients' symptoms, quality of life, or functional status. The known symptoms and interventions each patient experienced are presented separately in the results section. To emphasize this, we reiterated in the discussion section:

“One patient underwent arthrocentesis, another synovectomy, and yet another joint replacement.”

And to the limitations section:

“Due to the retrospective nature of the study, assessing the symptoms of the affected patients was limited to the data in the clinical records.”

Comment 5:

Selection bias and survival bias

As acknowledged by the authors, patients with shorter survival were unlikely to develop late arthropathy, resulting in survival bias.

However, this issue is not sufficiently addressed in the interpretation of the results: the conclusion currently states “There is a higher incidence…” which overstates the certainty.

Please modify this conclusion to reflect the limitations (e.g., “may be associated with higher risk,” or “was observed more frequently in this retrospective cohort, though subject to bias”).

Response:

We appreciate your careful review. In addition to adding CIF analysis to correct for survival bias, we removed the statement about higher incidence from the abstract and revised it to reflect the limitations.

“We found progressive arthropathy, mostly painless, in one or more joints or intervertebral spaces of patients receiving crizotinib for NSCLC.”

In the conclusion section, we changed the statement to:

“In this single-center retrospective study, we found that rapidly progressive arthropathy was observed more frequently in one or more joints of patients receiving crizotinib for NSCLC.”

Minor Comments

1. Figure legends are overly descriptive and read like clinical notes. Please revise for clarity and conciseness — e.g., “Representative CT images of progressive shoulder arthropathy after crizotinib treatment.” The legends currently include long lists of arrows (red arrow, yellow arrow, orange arrow- which are not seen until Fig 3) and serial descri

---

## [Decision Letter · Decision Letter 1]

26 Oct 2025

Dear Dr. Eshet,

Thank you for submitting your manuscript to PLOS ONE. After careful consideration, we feel that it has merit but does not fully meet PLOS ONE’s publication criteria as it currently stands. Therefore, we invite you to submit a revised version of the manuscript that addresses the points raised during the review process.

The manuscript has been assessed by three reviewers, and their comments are appended below.

In addition, please address the following:

In your Methods section, you state that the image evaluators had access to identifying information. Could you please clarify what is meant by this?Related to the above, could you please confirm whether this was approved by your IRB, and whether informed consent was waived? It would be helpful to upload a full English translation of your IRB approval letter for the editor and reviewers to assess. This does not need to be an official translation.Given that identifying information was available, this can introduce bias by unblinding the image evaluators to the treatment group (crizotinib vs no crizotinib) during image assessment. Please acknowledge this limitation in your manuscript.Please report a confidence limit for the result that arthropathy was present in 18% of crizotinib-treated patients.Given the very small sample size for this study and the long period of observation, please take care not to overstate the conclusions. Please remove the sentence " There is a higher incidence of progressive arthropathy in NSCLC patients treated with crizotinib" from the abstract, or rephrase it to temper the conclusions given the setting and sample size of the study.In general, it would be advisable to be less conclusive with this study's findings throughout, given the limitations noted above.

Could you please carefully address these comments?

We look forward to receiving your revised manuscript.

Kind regards,

Alejandro Torrado Pacheco, PhD

Staff Editor

PLOS ONE

Journal Requirements:

Reviewers' comments:

Reviewer's Responses to Questions

**Comments to the Author**

Reviewer #1: All comments have been addressed

Reviewer #2: All comments have been addressed

Reviewer #3: All comments have been addressed

2. Is the manuscript technically sound, and do the data support the conclusions?

Reviewer #1: Yes

Reviewer #2: Yes

Reviewer #3: Yes

3. Has the statistical analysis been performed appropriately and rigorously?

Reviewer #1: Yes

Reviewer #2: Yes

Reviewer #3: Yes

4. Have the authors made all data underlying the findings in their manuscript fully available?

Reviewer #1: Yes

Reviewer #2: No

Reviewer #3: Yes

5. Is the manuscript presented in an intelligible fashion and written in standard English?

Reviewer #1: Yes

Reviewer #2: Yes

Reviewer #3: Yes

Reviewer #1: The corresponding author has adequately addressed the comments and effected the necessary changes in the manuscript. I recommend that the article be accepted for publication.

Reviewer #2: The revised manuscript satisfactorily addresses all major and minor comments from the previous round. The authors added detailed statistical methods, clarified image assessment procedures, restructured tables and legends, and tempered the conclusions to match the data limitations. The competing-risk analysis appropriately adjusts for follow-up bias, and the overall presentation is clear and professional.

One remaining limitation is that the data are not fully publicly available, as access is restricted via the institutional ethics committee. If ethically feasible, I encourage deposition of anonymized imaging-derived data in a public repository.

Otherwise, the manuscript is technically sound, statistically appropriate, and clearly written. I recommend acceptance after editorial verification of the data-availability compliance.

Reviewer #3: None for this revision.

**Do you want your identity to be public for this peer review?** For information about this choice, including consent withdrawal, please see our Privacy Policy

Reviewer #1: No

Reviewer #2: No

Reviewer #3: No

---

## [Author Response · Author response to Decision Letter 2]

3 Nov 2025

Dear Dr. Alejandro Torrado Pacheco,

Staff Editor

PLOS ONE

Thank you for the opportunity to submit a revised draft of our manuscript titled

“Rapidly progressive arthropathy identified on imaging of patients treated with

Crizotinib for ALK-rearranged/ROS1-positive non-small cell lung cancer: a

retrospective single-center study” to PLOS ONE.

We have incorporated changes to address the editors’ suggestions.

The changes have been highlighted within the manuscript.

Here is a point-by-point response to the editors’ comments and concerns.

Comment 1:

In your Methods section, you state that the image evaluators had access to identifying information. Could you please clarify what is meant by this?

Response:

Thank you for the chance to clarify this point. The image evaluators used the patient’s real identification number to locate the study on the picture archiving and communication system (PACS) system, and they could also see the dates of the study, but they were unaware of the patient’s clinical details, specifically whether the patients received Crizotinib or whether they were symptomatic. We changed in the text:

“The readers … were blinded they their clinical symptoms and whether they were treated with crizotinib”.

Comment 2:

Related to the above, could you please confirm whether this was approved by your IRB, and whether informed consent was waived? It would be helpful to upload a full English translation of your IRB approval letter for the editor and reviewers to assess. This does not need to be an official translation.

Response:

In the methods section it is stated:

“Approval from an institutional review board was obtained for the retrospective assessment of the patient’s clinical and radiological records. The patient’s informed consent was waived due to the study's retrospective nature.”

I have uploades a full English translation of our IRB approval letter, as well as confirmation of approval by the Institutional Review Board.

Comment 3:

Given that identifying information was available, this can introduce bias by unblinding the image evaluators to the treatment group (crizotinib vs no crizotinib) during image assessment. Please acknowledge this limitation in your manuscript.

Response:

The image evaluators were unaware of the treatment group during image assessment.

Comment 4:

Please report a confidence limit for the result that arthropathy was present in 18% of crizotinib-treated patients.

Response:

I added Confidence limit to the abstract, as it was already included in the results section.

Comments 5+6

Given the very small sample size for this study and the long period of observation, please take care not to overstate the conclusions. Please remove the sentence " There is a higher incidence of progressive arthropathy in NSCLC patients treated with crizotinib" from the abstract, or rephrase it to temper the conclusions given the setting and sample size of the study.

In general, it would be advisable to be less conclusive with this study's findings throughout, given the limitations noted above.

Response:

I sincerely apologize for any oversight in ensuring consistency across the manuscript and the editorial manager system. The sentence in question has already been revised in the updated version of the manuscript to: “We found progressive arthropathy, mostly painless, in one or more joints or intervertebral spaces of patients receiving crizotinib for NSCL”, but I forgot to change it in the editorial manager system.

I appreciate your thoughtful guidance regarding the importance of tempering conclusions, particularly given the small sample size and extended observation period. I will make sure to update the abstract within the editorial manager system to reflect the revised wording.

---

## [Editor Report · Decision Letter 2]

19 Dec 2025

Dear Dr. Eshet,

We look forward to receiving your revised manuscript.

Kind regards,

Alejandro Torrado Pacheco, PhD

Staff Editor

PLOS One
---

## [Author Response · Author response to Decision Letter 3]

21 Dec 2025

Dear Dr. Alejandro Torrado Pacheco,

Staff Editor

PLOS ONE

Thank you for the opportunity to submit a revised draft of our manuscript titled

“Rapidly progressive arthropathy identified on imaging of patients treated with

Crizotinib for ALK-rearranged/ROS1-positive non-small cell lung cancer: a

retrospective single-center study” to PLOS ONE.

We have incorporated changes to address the editors’ suggestions.

The changes have been highlighted within the manuscript.

Here is a point-by-point response to the editors’ comments and concerns.

Comment 1:

Discussion paragraph 1: "Such findings were not found in patients receiving other TKI inhibitors for NSCLC, implying that this is not a class effect but rather a drug-specific effect." > temper this statement as it cannot be supported by the data. At most this "may be" a drug-specific effect.

Response:

We changed in the text:

“Such findings were not found in patients receiving other TKI inhibitors for NSCLC, implying that this is not a class effect but rather may be a drug-specific effect.”

Comment 2:

Discussion last paragraph: "To conclude, we found progressive arthropathy..." > replace "adverse event" with "potential adverse event".

Response:

We changed in the text:

“To conclude, we found progressive arthropathy… and radiologists should be aware of this potential adverse event.”

---

## [Editor Report · Decision Letter 3]

15 Jan 2026

Rapidly progressive arthropathy identified on imaging of

patients treated with Crizotinib for ALK-rearranged/ROS1-positive non small cell lung cancer: a retrospective single-center study

PONE-D-25-06712R3

Dear Dr. Eshet,

We’re pleased to inform you that your manuscript has been judged scientifically suitable for publication and will be formally accepted for publication once it meets all outstanding technical requirements.

Kind regards,

James Mockridge

Staff Editor

PLOS One
---

## [Editor Report · Acceptance letter]

PONE-D-25-06712R3

PLOS One

Dear Dr. Eshet,

I'm pleased to inform you that your manuscript has been deemed suitable for publication in PLOS One. Congratulations! Your manuscript is now being handed over to our production team.

Kind regards,

on behalf of

Dr James Mockridge

Staff Editor

PLOS One